# Genome-wide identification of major genes and genomic prediction using high-density and text-mined gene-based SNP panels in Hanwoo (Korean cattle)

**Hyo Jun Lee**[1☯]**, Yoon Ji Chung**[1☯]**, Sungbong Jang**[2]**, Dong Won Seo**[1]**, Hak Kyo Lee**[3]**, Duhak Yoon**[4]**, Dajeong Lim**[5]*****, Seung Hwan Lee**[1]*****

**1** Division of Animal and Dairy Science, Chungnam National University, Daejeon, Korea, **2** Department of Animal and Dairy Science, University of Georgia, Athens, GA, United States of America, **3** Department of Animal Biotechnology, Chonbuk National University, Jeonju, Korea, **4** Department of Animal Science, Kyungpook National University, Sangju, Korea, **5** Animal Genome & Bioinformatics, National Institute of Animal Science, Wanju, Korea

☯ These authors contributed equally to this work.
* lim_dj@korea.kr (DL); slee46@cnu.ac.kr (SHL)

**Data Availability Statement:** Data cannot be shared publicly by the authors due to legal restrictions on data use agreement sharing

## Abstract

It was hypothesized that single-nucleotide polymorphisms (SNPs) extracted from text-mined genes could be more tightly related to causal variant for each trait and that differentially weighting of this SNP panel in the GBLUP model could improve the performance of genomic prediction in cattle. Fitting two GRMs constructed by text-mined SNPs and SNPs except text-mined SNPs from 777k SNPs set (exp_777K) as different random effects showed better accuracy than fitting one GRM (lm_777K) for six traits (e.g. backfat thickness: + 0.002, eye muscle area: + 0.014, Warner–Bratzler Shear Force of *semimembranosus* and *longissimus dorsi*: + 0.024 and + 0.068, intramuscular fat content of *semimembranosus* and *longissimus dorsi*: + 0.008 and + 0.018). These results can suggest that attempts to incorporate text mining into genomic predictions seem valuable, and further study using text mining can be expected to present the significant results.

## Introduction

Genomic prediction, which is the first step in genomic selection, is a method for calculating genomic estimated breeding values (GEBVs) using large numbers of genetic markers, such as single-nucleotide polymorphism (SNP), covering the whole genome [1]. The genomic prediction methods that are currently applied to livestock populations use the extent of linkage disequilibrium between markers and quantitative trait loci (QTL) because high-density SNPs increase the chances of co-segregation of markers with causal mutations [2]. Genetic variation in quantitative traits could be influenced by large numbers of loci affecting any given trait with small to moderate effects. In some cases, however, there are loci with moderate to large effects due to relatively recently selected mutations [3–5]. It is difficult to capture recently selected causal mutations in genomic prediction because the linkage disequilibrium between these

restrictions. All the data-set used in this study was provided by BioGreen 21 Program (Molecular Breeding Program) of National Institute of Animal Science, RDA. The carcass traits can be obtained at public web site (https://mtrace.go.kr/). Request for Genotype and meat quality traits data can be made to Korea National Institute of Animal Science, Animal Genome & Bioinformatics Division (http://www.nias.go.kr/english/sub/boardHtml.do?boardId=depintro), Tae Hun Kim, PhD, Director of Animal Genome & Bioinformatics Division, (thkim63@korea.kr). All other relevant data are within the paper and its Supporting Information files

**Funding:** This study was funded by awards from the Molecular Breeding program (Grant no. PJ0131692020) of the Next Generation BIOGREEN21 project of the National Institute of Animal Science, RDA, Republic of Korea. Hyo Jun Lee was also partly supported by Institute of Information & communications Technology Planning & Evaluation (IITP) grant funded by the Korea government (MSIT) (No.2020-0-01441, Artificial Intelligence Convergence Research Center (Chungnam National University)).

**Competing interests:** The authors have declared that no competing interests exist.

mutations and other markers is incomplete [6]. Therefore, it is necessary to understand the genetic processes and information related to quantitative or complex traits more fully, as well as linkage disequilibrium between causal variants and common SNPs, to increase the ability of genomic prediction models. Genomic best linear unbiased prediction (GBLUP) is a commonly used method that has been widely utilized for genomic prediction. The main assumption of the GBLUP method is that most SNPs have small effects with a normal distribution, regardless of prior biological information on the genetic architecture of the traits [7]. However, the effects of SNPs associated with quantitative traits are not always normally distributed and the effects may differ depending on the biological processes of the traits. For these reasons, it might be necessary to incorporate previous biological knowledge into the GBLUP method for more accurate genomic prediction. In previous studies, when selected SNP panels based on biological information were weighted differentially in the GBLUP method, higher prediction accuracy was obtained compared with the normal GBLUP [8, 9]. In addition, using causal genes or markers with prior biological knowledge resulted in much more accurate QTL discovery [10].

As mentioned in the paragraph above, it is necessary to understand the biological characteristics of complex traits from previous studies for more accurate genomic prediction. However, manually scanning previous studies to analyze biological information requires a lot of time and effort because there are many published studies in the field of animal science, and the number is expanding at an increasing rate. As of 2018, approximately 29 million papers were cited in PubMed, one of the most commonly used life science databases (https://www.nlm.nih.gov/bsd/licensee/baselinestats.html). In addition, the majority of published papers are composed of unstructured text, which is difficult to use for other studies. Therefore, it is important to use techniques to extract useful information from the textual data without spending a lot of time. Text mining is one technique for resolving this problem [11]. In the biomedical field, text mining has been used to assist studies in gene–disease associations and gene–gene associations, and to analyze clinical datasets to improve quality of health care [12–14]. In addition, text mining has been widely applied in various fields other than biomedicine, such as business and marketing [15]. However, in the field of animal breeding, studies using text mining are still rare. The application of text mining to genomic prediction could be an interesting approach to animal breeding studies. In this study, text mining was used to identify genes associated with carcass and meat quality traits, and these text-mined genes with biological information were used for genomic prediction. The hypothesis of this study was that SNPs extracted from text-mined genes could be in tighter linkage disequilibrium with causal variants for carcass and meat quality traits, and weighting this SNP panel differentially in the GBLUP model could improve the performance of genomic prediction in cattle.

## Materials and methods

### Dataset

**Hanwoo (Korean cattle) populations.** The Animal Care and Use Committee of the National Institute of Animal Science (NIAS), Rural Development Administration (RDA), South Korea, approved the experimental procedures, and appropriate animal health and welfare guidelines were followed. The Hanwoo were sourced from two different commercial populations based on different phenotype measurements. The first commercial population included 12,635 individuals (animals were born between 2013 and 2016 and samples were collected between 2017 and 2019) evaluated for carcass traits (CWT, EMA, and BF). The second population consisted of 1,039 steers evaluated for meat quality traits (Warner–Bratzler Shear Force [WBSF] and intramuscular fat content). The two populations were half-sibs derived from 339 sires for the first population and 82 sires for the second population, with unrelated

dams. All animals of the two populations ($n$ = 12,635, $n$ = 1,039) were slaughtered at averages of 918 and 920 days, respectively. The carcass traits ($n$ = 12,635) consisted of three traits. The carcass weight (CWT/kg), backfat thickness (BF/mm), and eye muscle area (EMA/cm$^2$) were measured after a 24-hour chill at the junction of the 12th and 13th ribs. Meat quality traits ($n$ = 1,039) were measured by evaluating two traits in two muscles. The WBSF values of the longissimus dorsi muscle (D_SF) and semimembranosus muscle (S_SF) were measured according to the method described by Wheeler et al. (2000) [16]. Briefly, beef steak 2.5 cm$^2$ thick was kept in polyethylene bags for 48 hours postmortem. All of the bags were heated in a water bath at 80°C for 30 minutes, until the internal temperature of the steaks reached 70°C. The samples were stored at room temperature for 30 minutes prior to measurement. An Instron Universal WBSF testing machine (Instron Corporation, Canton, MA) with a cross-head speed of 200 mm/min and a 50-kg load cell was used to measure the WBSF. Each sample was divided into six representative cores with a diameter of 1.27 cm and parallel to the muscle fibers. The final phenotype of the WBSF was the mean of the maximum force required to shear each core sample. The intramuscular fat contents of the longissimus dorsi muscle (D_IMF) and semimembranosus muscle (S_IMF) were measured using the microwave solvent extraction method described by AOAC International [17].

**Genotyping and quality control.** The genomic DNA of each animal group was extracted from longissimus thoracis muscle samples using a DNeasy Blood and Tissue Kit (Qiagen, Valencia, CA). DNA concentration and purity were determined using a NanoDrop 1000 (Thermo Fisher Scientific, Wilmington, DE). A total of 13,674 samples were genotyped using the Illumina Bovine SNP50 BeadChip and the 1,295 samples were genotyped additionally by the Illumina Bovine HD BeadChip to use as the reference population in imputation step. All animals' 50K genotypes were imputed to a high density level (777K) using Minimac3 [18]. $r^2 < 0.6$ SNPs were excluded in the imputations step and SNPs on the sex chromosomes were excluded from the analysis. SNP quality control for each group was performed using PLINK1.9 software [19] based on the following criteria: minor allele frequency < 0.001 for carcass traits group and < 0.01 for meat quality traits group; gene call rate < 0.1. In the carcass trait group, 23,415 SNPs were excluded by the above step, leaving 670,080 SNPs. In the meat quality trait group, 56,477 SNPs were excluded by this step, and 637,017 SNPs were used for the analysis. The imputed 777K SNPs of each group were annotated using the SnpEff program [20].

**Text mining and gene ontology term analysis.** Published papers related to CWT, WBSF, IMF, BF, and EMA were searched before text mining. The workflow of the text mining is shown in S1 Fig. First, all the texts in the abstracts of papers containing queries related to traits in their abstracts or titles were collected. This step was performed using functions in the RISmed package of the R statistical programming language [21]. Words consisting of only capital letters or numbers were extracted to filter out words that were accidentally the same as gene symbols (*e.g.*, impact, pigs). Finally, only words matching the bovine gene symbols in the BioMart databases were selected for analysis. The gene symbols were obtained from the Bioconductor package BiomaRT, and btaurus_gene_ensembl was used as the dataset [22]. SNPs contained in text-mined genes (TMG) were then extracted from the imputed 777K SNPs. Furthermore, SNPs from the intergenic region of TMG were also extracted because the intergenic region often contains functionally important elements, such as promoters and enhancers. The above two types of SNPs were used as text-mined SNPs. In this study, three marker sets—the imputed 777K SNPs (Im_777K), the SNPs excluding the text-mined SNPs from imputed 777K SNPs (exp_777K), and the text-mined SNPs—were used in genomic prediction. The Bioconductor R package 'clusterProfiler' was used for Gene Ontology (GO) analysis to identify the biological process of TMG [23]. The $-\log_{10}$ adjusted *P*-value (P.adj) by the Bonferroni method

was used to examine the significance in GO analysis. To visualize the differences between QTL regions obtained from Animal QTL DB [24] and text-mined regions, karyotypes were plotted using the Circos program [25].

## Statistical analyses

**Genome-wide association study (GWAS) using text-mined gene-based SNP panels.** The phenotypic data on carcass and meat quality traits were pre-adjusted for fixed effects including growing sites, birth year, season, and slaughter age using a linear model implemented in R software 3.3.1 (R Foundation for Statistical Computing, Vienna, Austria). The adjusted phenotypes and text-mined SNP panel were subsequently used for GWAS under a linear mixed model. The linear mixed model can be written as:

$$\mathbf{y}_c = \mu \mathbf{1_N} + \mathbf{D\beta} + \mathbf{g} + \mathbf{e}$$

where $\mathbf{y}_c$ is a vector of the corrected phenotype for N individuals; $\mu$ is the overall mean of the term and $\mathbf{1_N}$ is a vector of N ones; $\mathbf{D}$ is a vector of genotype of the candidate SNPs recorded as 0, 1, or 2; $\mathbf{\beta}$ is the additive effect of the candidate SNPs; $\mathbf{g}$ is a vector of random polygenic effects from the genetic relationship matrix (GRM) constructed by the lm_777K; and $\mathbf{e}$ is a vector of residuals. This model was computed by GCTA 1.26 [26]. The GRM for the polygenic effect ($\mathbf{g}$) was constructed using all SNPs except those on the chromosome where the candidate SNP was located. The *P*-values were adjusted using the Bonferroni method to correct multiple hypotheses. The values calculated by dividing 0.05 by the number of text-mined SNPs were used as the thresholds for obtaining significant SNPs associated with the trait.

**Genomic models for estimation and prediction.** The three genomic models were used to estimate genetic and residual variances as well as to predict genomic estimated breeding values (GEBV) in models 1 to 3. The two types of GRM constructed by lm_777K and exp_777K were used for models 1 and 2, respectively. The equations can be written as:

$$\mathbf{y} = \mu \mathbf{1_N} + \mathbf{Xb} + \mathbf{g_{all}} + \mathbf{e} \qquad \text{(model 1)}$$

$$\mathbf{y} = \mu \mathbf{1_N} + \mathbf{Xb} + \mathbf{g_{-t}} + \mathbf{e} \qquad \text{(model 2)}$$

where $\mathbf{y}$ is the vector of the observed phenotype for N individuals. $\mathbf{X}$ is an incidence matrix for the fixed effects and $\mathbf{b}$ is the vector of fixed effects, which included growing site, birth month, birth year, slaughter month, slaughter year, and slaughter age as covariates for all traits. In addition, the carcass traits included slaughter place and sex, while the meat quality trait included farm information (the owner's name of steers). In the two equations, $\mathbf{g_{all}}$ is the N vector of the additive effects from the GRM with lm_777K for additive genetic effects, and $\mathbf{g_{-t}}$ is the N vector of the additive effects from the GRM with exp_777K. The genetic and residual effects were assumed to be normally distributed, with mean as zero. The variances estimated by the above two models are given by:

$$\text{Var}\begin{bmatrix} \mathbf{g_{all}} \\ \mathbf{e} \end{bmatrix} = \begin{bmatrix} \mathbf{G_{all}}\sigma^2_{all} & 0 \\ 0 & \mathbf{I}\sigma^2_E \end{bmatrix}, \; \text{Var}\begin{bmatrix} \mathbf{g_{-t}} \\ \mathbf{e} \end{bmatrix} = \begin{bmatrix} \mathbf{G_{-t}}\sigma^2_{-t} & 0 \\ 0 & \mathbf{I}\sigma^2_E \end{bmatrix}$$

where $\mathbf{G_{all}}$ and $\mathbf{G_{-t}}$ are GRMs with lm_777K and exp_777K, respectively; and $\mathbf{I}$ is an N*N identity matrix.

In model 3, two GRMs constructed by exp_777K and text-mined SNPs were jointly used to differentially weight the random effects. The model used can be written as:

$$\mathbf{y} = \mu \mathbf{1_N} + \mathbf{Xb} + \mathbf{g_{-t}} + \mathbf{g_t} + \mathbf{e} \qquad \text{(model 3)}$$

where **y** is the vector of phenotypic observations, and $\mathbf{g_t}$ is the N vector of the additive effects from GRM with the text-mined SNPs. The genetic and residual effects were assumed to be normally distributed, with mean as zero. The variances estimated by model 3 are given by:

$$\text{Var}\begin{bmatrix} \mathbf{g_t} \\ \mathbf{g_{-t}} \\ \mathbf{e} \end{bmatrix} = \begin{bmatrix} \mathbf{G_t}\sigma_t^2 & 0 & 0 \\ 0 & \mathbf{G_{-t}}\sigma_{-t}^2 & 0 \\ 0 & 0 & \mathbf{I}\sigma_E^2 \end{bmatrix}$$

where $\mathbf{G_t}$ is the GRM with the text-mined SNPs.

**Variance component estimation and GBLUP.** The variance components, $\sigma_{all}^2$, $\sigma_{-t}^2$, and $\sigma_t^2$, and heritability were estimated using an average information restricted maximum likelihood (AIREML) model by implementing the AIREMLF90 program in the BLUPF90 family [27]. The proportion of genomic variance explained by each model can be written as:

$$h^2 = \frac{\sigma_{all}^2}{\sigma_{all}^2 + e} \qquad \text{(model 1)}$$

$$h^2 = \frac{\sigma_{-t}^2}{\sigma_{-t}^2 + e} \qquad \text{(model 2)}$$

$$h^2 = \frac{\sigma_t^2 + \sigma_{-t}^2}{\sigma_t^2 + \sigma_{-t}^2 + e} \qquad \text{(model 3)}$$

GEBVs were predicted using GBLUP methods and a 10-fold cross-validation scheme was used to evaluate the accuracy of the GEBVs. Samples were divided into 10 groups of equal size. Nine of these groups were used as the reference set and the other group was used as the validation set in each cross-validation. The GEBVs for the model 1 and model 2 were calculated using the following mixed model. The matrix for the model used can be written as:

$$\begin{bmatrix} b \\ \hat{u} \end{bmatrix} = \begin{bmatrix} X'X & X'Z \\ Z'X & Z'Z + \lambda G^{-1} \end{bmatrix}^{-1} \begin{bmatrix} X'y \\ Z'y \end{bmatrix}$$

where $\hat{u}$ is the vector of the GEBVs distributed as g~(0,$G\sigma_g^2$); $G$ is genomic relationship matrix for individuals; $Z$ is a design matrix designed one column for each GEBV and one row for each phenotype (if an individual would have no phenotype, $Z$ would have a column with zero's only for this individual). $\lambda$ is shrinkage value calculated by ($\sigma_e^2$ /$\sigma_g^2$). The GEBV for the model 3 is calculated using two random effect linear mixed model followed by

$$\begin{bmatrix} b \\ \hat{u}_{-t} \\ \hat{u}_t \end{bmatrix} = \begin{bmatrix} X'X & X'Z & X'Z \\ Z'X & Z'Z + \lambda^{-t}G_{-t}^{-1} & Z'Z \\ Z'X & Z'Z & Z'Z + \lambda^t G_t^{-1} \end{bmatrix}^{-1} \begin{bmatrix} X'y \\ Z'y \\ Z'y \end{bmatrix}$$

Where $\hat{u}_{-t}$ and $\hat{u}_t$ are vectors of GEBVs calculated by exp_777K and text-mined SNPs; $G_{-t}$ and $G_t$ are GRMs with exp_777k and text-mined SNPs. The final GEBV of model 3 is the sum of the two GEBVs ($\hat{u}_{-t} + \hat{u}_t$). The GRM ($G$) is defined as

$$\mathbf{G} = \frac{\mathbf{MM'}}{2\sum p_j(1 - p_j)}$$

where **M** contains genotypes adjusted by allele frequency and $p_j$ is the allele frequency for

marker j [28]. All of these estimates were performed using BLUPF90 [27]. The accuracy of predicted breeding values was calculated as the Pearson's correlation between the GEBVs and adjusted phenotypes ($y_c$) of the validation set, and the equation can be represented by:

$$\text{Accuracy} = r_{(GEBV, y_c)}$$

## Results

### Text mining and gene ontology term analysis

The queries used to search the papers and a statistical summary of the text mining are shown in Table 1. Regarding number of searched articles, CWT ranked first with 1893 papers, followed with IMF, WBSF, BF, EMA with (1854, 1097, 602, 546), respectively. In the number of calling genes, IMF showed the largest number of genes with 576, although a similar number of papers with CWT were searched. Other traits were ranked in order of CWT, BF, EMA, WBSF with (288, 195, 167, 156). The 30 genes that appeared with highest frequency in text mining are shown in Table 2. The most matched gene to bovine gene symbols in each trait were (CWT: *IGF1*(36 times), WBSF: *CAST*(110 times), IMF: *SCD*(105 times), BF: *MC4R*(35 times), and EMA: *MSTN*(19 times)), respectively.

In the results of Gene Ontology (GO) term analysis (Table 3), CWT, BF, EMA-related TMG showed significance relatedness with growth regulator and growth factor ("response to hormone", "regulation of signaling receptor activity", and "response to endogenous stimulus", "response to peptide"). WBSF-related TMG were identified to be associated with organic acid ("carboxylic acid metabolic process", "oxoacid metabolic process", "monocarboxylic acid biosynthetic process", "organic acid metabolic process", "monocarboxylic acid metabolic process"). For IMF, the biological process terms with lipid synthesis and lipid metabolism were statistically significant ("regulation of lipid metabolic process", "lipid metabolic process", "fatty acid metabolic process", "regulation of lipid biosynthetic process"). The karyotypes of the QTL regions registered in animal QTLDB, text-mined regions, and the intersection of the two regions are shown in Fig 1. The highest percentage of intersecting regions within the text-mined regions corresponded to regions of CWT-related TMG (36.3%), and the lowest corresponded to IMF regions (5.5%).

### Genome-wide association study (GWAS) with text-mined SNPs

The Manhattan plots for each trait are shown in Fig 2. The Bonferroni correction method was used for the significance test (0.05/number of SNPs) in the genome-wide association study,

**Table 1. Summary statistics of text mining and SNP calling.**

| Trait | Article[6] | Gene[7] | SNP[8] | Used query[9] |
|---|---|---|---|---|
| **CWT[1]** | 1,893 | 288 | 17,662 | carcass weight[TIAB] OR dressed weight[TIAB] |
| **WBSF[2]** | 1,097 | 156 | 6,143 | Warner-Bratzler Shear Force [TIAB] OR cuttability [TIAB] OR meat tenderness [TIAB] |
| **IMF[3]** | 1,854 | 576 | 30,983 | intramuscular fat [TIAB] |
| **BF[4]** | 602 | 195 | 9,335 | back fat [TIAB] |
| **EMA[5]** | 546 | 167 | 12,371 | eye muscle area [TIAB] OR ribeye [TIAB] OR rib eye [TIAB] |

**CWT[1]**: Carcass weight; **SF[2]**: Warner-Bratzler Shear Force; **IMF[3]**: intramuscular fatty acid content; **BF[4]**: Backfat thickness; **EMA[5]**: Eye muscle area; **Article[6]**: number of articles searched in PubMed; **Gene[7]**: number of mined genes from searched articles; **SNP[8]**: number of SNPs called from imputed 777K markers; **Used query[9]**: queries used to search articles in PubMed.

and the SnpEff annotation information was referenced for marker locations. Three significant clusters were found in CWT. The most significant markers at position 10710350 in chromosome 4 are involved in the intron region of *CALCR* gene (P = $10^{-29.6}$). In the genomic region of chromosomes 6 and 14, markers involved in the *LCORL–SLIT2* (position: 39,932,557) and *PLAG1–CHCHD7* (position: 25,015,640) intergenic regions showed the most significance (P = $10^{-40.2}$ and P = $10^{-105.3}$). There are four significant genomic regions in BF. The most significant marker on chromosome 22 is located at a downstream gene variant of *PPARG* (position: 57,362,666; P = $10^{-6.5}$). The other most significant markers in chromosomes 2, 13, and 23 clusters are located in *INSIG2–EN1* (position: 70,895,063; P = $10^{-5.97}$), *APCDD1L–VAPB* (position: 58,449,824; P = $10^{-7.3}$), and *BMP5–HMGCLL1* (position: 4,622,146; P = $10^{-16.9}$) intergenic region. Three clusters showed significance in EMA. The most significant markers in chromosomes 3, 6, and 14 are involved in the *S100A10–THEM4* (position: 18,822,190; P = $10^{-6.2}$), *LCORL–SLIT2* (position: 39,932,557; P = $10^{-12.5}$), and *PLAG1–CHCHD7* (position: 25,015,640; P = $10^{-26.6}$). For meat quality traits, only one marker at position 98,540,675 on chromosome 7 showed significance for D_SF (P = $10^{-7.4}$), located in an intron variant of the *CAST* gene.

**Table 2. The 30 genes symbol that appeared with highest frequency in text mining.**

| Trait | Symbol | Freq | Trait | Symbol | Freq | Trait | Symbol | Freq |
|---|---|---|---|---|---|---|---|---|
| CWT | IGF1 | 36 | BF | MC4R | 35 | EMA | MSTN | 19 |
| | MSTN | 28 | | SST | 26 | | CAPN1 | 18 |
| | MC4R | 25 | | IGF1 | 24 | | ADIPOQ | 15 |
| | LPL | 24 | | FTO | 18 | | LEPR | 15 |
| | TNF | 24 | | GAA | 16 | | PPARGC1A | 15 |
| | BLM | 20 | | SLA | 15 | | DES | 13 |
| | CAPN1 | 19 | | FASN | 14 | | POMC | 13 |
| | IGFBP2 | 19 | | IGF2 | 14 | | GHR | 12 |
| | MGA | 19 | | BSG | 11 | | LEP | 12 |
| | NCAPG | 19 | | MGA | 11 | | PIK3C3 | 11 |
| | POMC | 18 | | RBP4 | 10 | | SLA | 11 |
| | IGF2 | 17 | | UCP2 | 10 | | CAST | 9 |
| | GHR | 16 | | SPR | 9 | | GH1 | 9 |
| | AFP | 15 | | CSTB | 8 | | IGF2 | 9 |
| | CRH | 13 | | FABP3 | 8 | | LRIT3 | 9 |
| | DGAT1 | 13 | | IGFBP3 | 8 | | LCORL | 8 |
| | FASN | 13 | | LSR | 8 | | MC4R | 8 |
| | GAA | 13 | | MAP2K6 | 8 | | RPE | 8 |
| | LCORL | 13 | | MTTP | 8 | | ANGPTL3 | 7 |
| | TRH | 13 | | SCD | 8 | | CRH | 7 |
| | CAPN3 | 11 | | STAT6 | 8 | | FABP4 | 7 |
| | CAST | 11 | | TNF | 8 | | GRP | 7 |
| | SCD | 11 | | CTSL | 7 | | MAP2K6 | 7 |
| | ABHD5 | 10 | | EZH2 | 7 | | AGAP3 | 6 |
| | ASL | 10 | | IRS4 | 7 | | BPI | 6 |
| | GNAS | 10 | | MARK4 | 7 | | ERG | 6 |
| | IGFBP3 | 10 | | QSOX1 | 7 | | IGF1 | 6 |
| | IGFBP4 | 10 | | SLC13A5 | 7 | | ADRB3 | 5 |
| | IRS1 | 10 | | TGFBR1 | 7 | | EMD | 5 |
| | STAT6 | 10 | | UCP3 | 7 | | ME1 | 5 |

*(Continued)*

**Table 2.** (Continued)

| Trait | Symbol | Freq | Trait | Symbol | Freq | Trait | Symbol | Freq |
|---|---|---|---|---|---|---|---|---|
| WBSF | CAST | 110 | IMF | SCD | 105 | | | |
| | CAPN1 | 104 | | LPL | 80 | | | |
| | CAPN3 | 19 | | FABP4 | 70 | | | |
| | KCNJ11 | 18 | | FAS | 54 | | | |
| | NES | 17 | | FABP3 | 52 | | | |
| | DNAJA1 | 16 | | FASN | 52 | | | |
| | MSTN | 14 | | LEPR | 47 | | | |
| | ADAMTS4 | 11 | | PPARG | 38 | | | |
| | DGAT1 | 11 | | DGAT1 | 36 | | | |
| | HSPB1 | 9 | | MC4R | 36 | | | |
| | SCD | 8 | | AFP | 27 | | | |
| | TNNT3 | 8 | | MSC | 26 | | | |
| | UCP3 | 8 | | CAST | 25 | | | |
| | ANGPTL3 | 7 | | PRKAG3 | 23 | | | |
| | IGFBP2 | 7 | | FTO | 22 | | | |
| | ADAMTS5 | 6 | | SREBF1 | 22 | | | |
| | CAPN2 | 6 | | CAPN1 | 20 | | | |
| | DLK1 | 6 | | MAT2B | 19 | | | |
| | MYOD1 | 6 | | PLIN2 | 17 | | | |
| | PRKAG3 | 6 | | RYR1 | 16 | | | |
| | STAT6 | 6 | | KLF6 | 15 | | | |
| | UCP2 | 6 | | ACACA | 14 | | | |
| | LEP | 5 | | ADH1C | 14 | | | |
| | MMP2 | 5 | | GPAM | 14 | | | |
| | APP | 4 | | IGF2 | 14 | | | |
| | FABP4 | 4 | | PDHB | 14 | | | |
| | GEN1 | 4 | | PPARA | 14 | | | |
| | IGF2 | 4 | | ASIP | 13 | | | |
| | LOX | 4 | | MSTN | 13 | | | |
| | MAP3K5 | 4 | | VRTN | 13 | | | |

## Variance component estimation

A statistical summary of the variance component estimation is shown in Table 4. In carcass traits, CWT showed the highest heritability (0.42) when Im_777K was used in the estimation. BF and EMA showed no difference in heritability between the three different estimation models (BF: 0.41, EMA: 0.39). In meat quality traits, the heritabilities of WBSF in the two muscle types *semimembranosus* and *longissimus dorsi* were 0.1 and 0.19, respectively, when estimated using the Im_777K panel. S_IMF and D_IMF showed heritabilities of 0.21 and 0.32, respectively, when estimated using Im_777K. All four traits showed similar heritabilities between the three models.

## Genomic prediction

The accuracy of GEBV are shown separately for the carcass traits (CWT, BF, EMA) and meat quality traits (WBSF, IMF) in Table 5. Fitting two different GRMs constructed with two different SNP panels (exp_777K + tm_SNPs) as random effects in the GBLUP model showed better accuracy than fitting one GRM with exp_777K in all traits. In CWT, the prediction accuracy

**Table 3. The top five significant biological processes for each trait.**

| Trait | GO_ID | Biological process | GeneRatio[1] | $-log_{10}P.adj$[2]: |
|-------|-------|--------------------|----------|-----------------|
| CWT | GO:0009725 | response to hormone | 19.8% | 9.5 |
| | GO:0010469 | regulation of signaling receptor activity | 21.4% | 8.2 |
| | GO:0009719 | response to endogenous stimulus | 24.6% | 7.5 |
| | GO:0043066 | negative regulation of apoptotic process | 19.0% | 6.9 |
| | GO:0043069 | negative regulation of programmed cell death | 19.0% | 6.7 |
| WBSF | GO:0019752 | carboxylic acid metabolic process | 21.7% | 2.6 |
| | GO:0043436 | oxoacid metabolic process | 21.7% | 2.4 |
| | GO:0072330 | monocarboxylic acid biosynthetic process | 12.0% | 2.3 |
| | GO:0006082 | organic acid metabolic process | 21.7% | 2.3 |
| | GO:0032787 | monocarboxylic acid metabolic process | 14.5% | 1.8 |
| IMF | GO:0019216 | regulation of lipid metabolic process | 11.5% | 12.7 |
| | GO:0032787 | monocarboxylic acid metabolic process | 15.3% | 12.2 |
| | GO:0006629 | lipid metabolic process | 23.4% | 11.5 |
| | GO:0006631 | fatty acid metabolic process | 11.1% | 9.4 |
| | GO:0046890 | regulation of lipid biosynthetic process | 7.7% | 9.3 |
| BF | GO:0009725 | response to hormone | 23.4% | 9.6 |
| | GO:0032868 | response to insulin | 12.8% | 8.2 |
| | GO:1901700 | response to oxygen-containing compound | 28.7% | 8.1 |
| | GO:0009719 | response to endogenous stimulus | 28.7% | 8.0 |
| | GO:0043434 | response to peptide hormone | 12.8% | 5.7 |
| EMA | GO:1901652 | response to peptide | 14.1% | 4.1 |
| | GO:0032868 | response to insulin | 11.3% | 4.0 |
| | GO:0010243 | response to organonitrogen compound | 19.7% | 4.0 |
| | GO:0043434 | response to peptide hormone | 12.7% | 3.6 |
| | GO:0062013 | positive regulation of small molecule metabolic process | 9.9% | 3.5 |

**GeneRatio[1]**: gene calling rate, i.e., the ratio of genes involved in each biological process among entire set of text-mined genes; $-log_{10}P.adj$[2]: −log10 P-value adjusted by the Bonferroni method.

with Im_777K was 0.453, which was 0.002 higher than in the model with exp_777K + tm_SNPs. Conversely, for BF, using exp_777K + tm_SNPs resulted in an accuracy of 0.421, which was 0.002 higher than that using Im_777K. EMA also exhibited its highest prediction accuracy (0.437) when using two GRMs with exp_777K + tm_SNPs. The accuracy of genomic prediction using two GRMs for WBSF in the two muscle types, *semimembranosus* and *longissimus dorsi*, were calculated as 0.129 and 0.189, respectively, and those for IMF were 0.168 and 0.225, respectively, which were better than those using Im_777K. In order to validate the effect of text-mined SNPs in the multi-GRM model, GBLUP using evenly-mined SNPs (em_SNPs) and except SNPs was additionally conducted (Table 6). For all four meat quality traits, the GBLUP using tm_SNPs showed higher accuracy than em_SNPs. It seems that CWT and EMA may have more polygenic characteristics than other traits, because em_SNPs showed higher accuracy than tm_SNPs in these two traits.

## Discussion

### Biological relatedness of text-mined gene with carcass and meat quality traits

**Carcass traits.** The top three mined genes for carcass traits were *IGF1, MSTN, MC4R, SST, CAPN1, and PPARGC1A*. Many previous studies have investigated the biological effect of

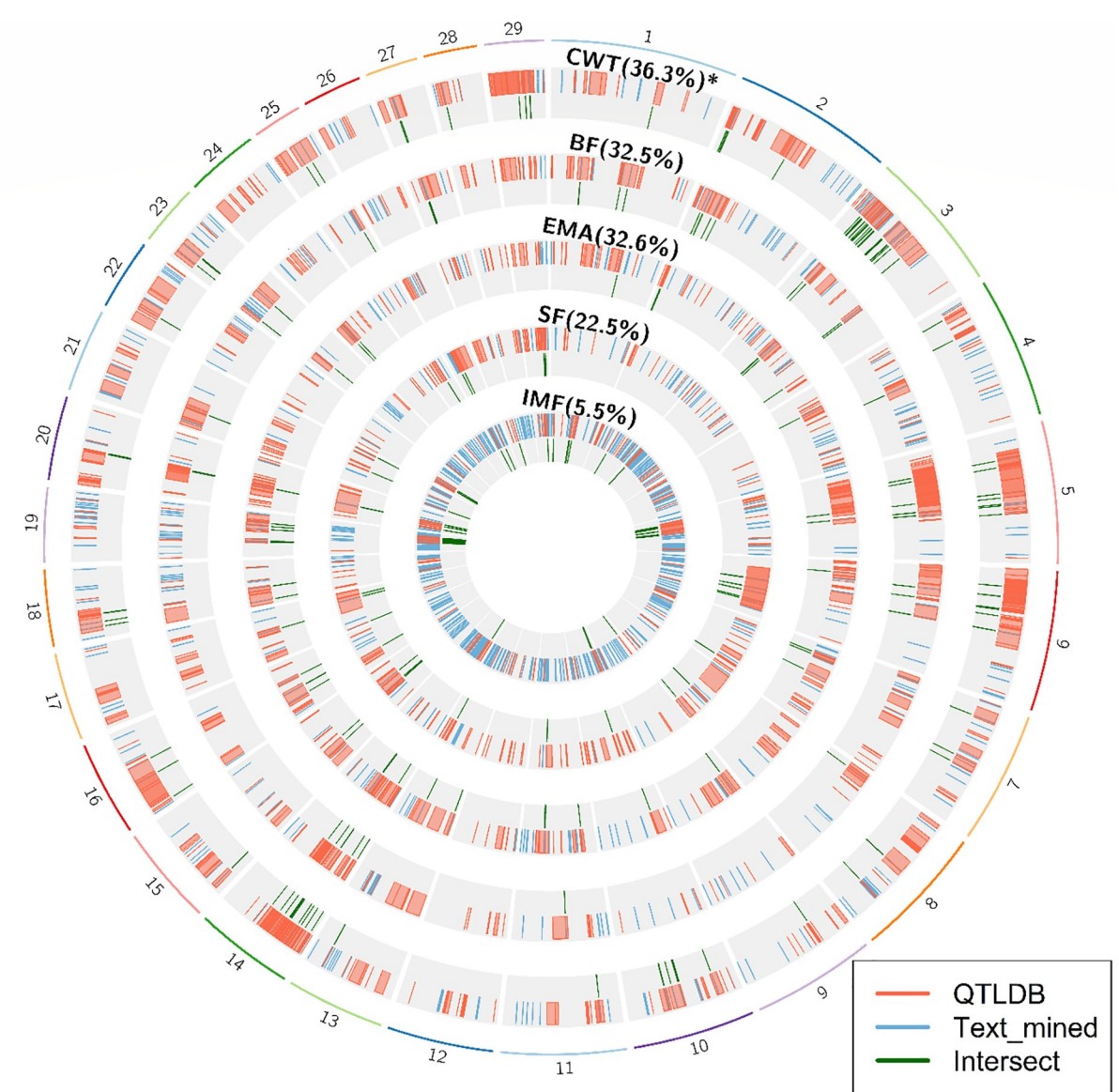

**Fig 1. The karyotype of QTL regions registered in QTLDB, text-mined region, and the intersection of both regions.** Each karyotype represents the region for the trait indicated above. Percentages in parentheses beside the trait names indicate the ratio of text-mined region within QTLDB region.

these genes on the quantitative traits. Insulin-like growth factor (*IGF*) plays a key role in cell differentiation, growth, and metabolism regulation [29]. The myostatin (*MSTN*) gene, also known as *GDF8*, encodes a member of the transforming growth factor β superfamily, which is associated with the proper regulation of skeletal muscle mass and carcass yield in cattle [30]. The melanocortin 4 receptor (*MC4R*) gene plays an important role in energy balance and is associated with beef economic traits [31]. Peroxisome proliferator activated receptor gamma coactivator 1 alpha (*PPARGC1A*) have been standing out as a candidate gene for beef fat synthesis [32]. Although somatostatin (*SST*) inhibits growth hormone, there has been little research on the association between the SST gene and carcass traits. This gene seemed to have been mined because the abbreviation "SST" was used with other meanings, such as "sole soft tissue", in the literature.

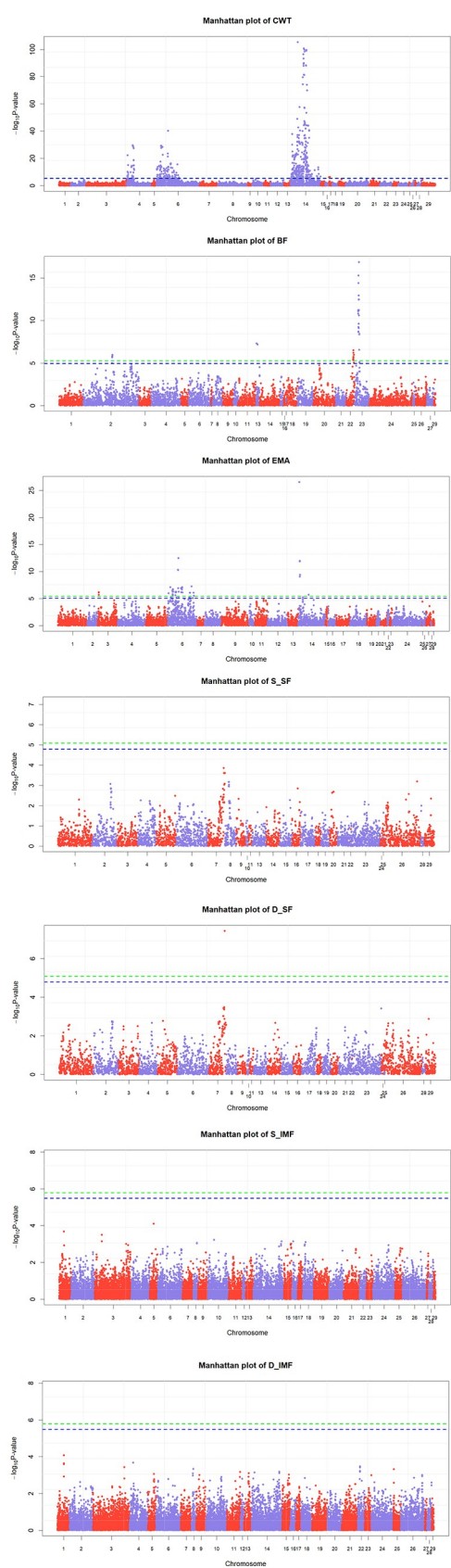

**Fig 2. Manhattan plots with results of genome-wide association study using text-mined SNPs for each trait.** The y-axis shows the $-\log_{10}P$-value of each SNP and the x-axis is the marker index. The green line is the Bonferroni-line representing 0.05/number of markers. The blue line is the suggestive-line representing 0.1/number of markers.

In addition to these high ranked genes, other genes (*i.e.*, *NCAPG*, *POMC*, *LCORL*, *FTO*, *IGF2*, *FABP3*, *LEPR*, and *ADIPOQ*) were also found to be associated with growth-related traits in multiple breed [33–40]. The significant genes in GWAS results (*CALCR*, *PLAG1*, *INSIG2*, *PPARG*, *BMP5*, *S100A10*) also have been identified to have relationship with growth performance and obesity of adipose tissue for pig and cattle [35, 41–45]. In addition, many other TMG also seems to be associated with growth related traits because the GO term results revealed that carcass traits-related TMG were associated with growth regulator and growth factor.

**Meat quality traits.** The *CAST* and *CAPN1* were included in the two most frequently mined genes related to the WBSF. Calpain 1 (*CAPN1*) encodes the large subunit of calcium-activated neutral proteases (calpain), and the calpastatin (*CAST*) gene inhibits μ- and m-calpain activity. These two proteins, as key myofibrillar proteins, mediate proteolysis during postmortem storage of the carcass and cuts of meat at refrigerated temperatures and play important roles in meat tenderness [46]. The association between these *CAST/CAPN1* and WBSF has been studied extensively [47–50]. In IMF, *SCD*, *LPL*, and *FABP4* were the three most frequently mined genes. The stearoyl-CoA desaturase (*SCD*) gene encodes an enzyme involved in fatty acid biosynthesis, primarily the synthesis of oleic acid [51]. The lipoprotein

**Table 4. Variance components at different marker set.**

| Trait | Value | Im_777K[1] | exp_777K[2] | exp_777K + tm_SNPs[3] |
|-------|-------|-----------|-------------|----------------------|
| CWT | $\sigma^2_u$ | 913.66 | 908.35 | 705.05 + 171.76 |
| | $\sigma^2_e$ | 1287.6 | 1297.2 | 1307.3 |
| | $h^2$ | 0.42 | 0.41 | 0.4 |
| BF | $\sigma^2_u$ | 9.51 | 9.44 | 8.91 + 0.63 |
| | $\sigma^2_e$ | 13.65 | 13.71 | 13.64 |
| | $h^2$ | 0.41 | 0.41 | 0.41 |
| EMA | $\sigma^2_u$ | 50.37 | 50.04 | 48 + 2.43 |
| | $\sigma^2_e$ | 77.59 | 77.87 | 77.55 |
| | $h^2$ | 0.39 | 0.39 | 0.39 |
| S_SF | $\sigma^2_u$ | 0.11 | 0.11 | 0.07 + 0.04 |
| | $\sigma^2_e$ | 1.02 | 1.02 | 1.02 |
| | $h^2$ | 0.1 | 0.09 | 0.09 |
| D_SF | $\sigma^2_u$ | 0.13 | 0.12 | 0.07 + 0.04 |
| | $\sigma^2_e$ | 0.55 | 0.55 | 0.55 |
| | $h^2$ | 0.19 | 0.18 | 0.17 |
| S_IMF | $\sigma^2_u$ | 0.66 | 0.67 | 0.65+ 0.000024 |
| | $\sigma^2_e$ | 2.46 | 2.44 | 2.47 |
| | $h^2$ | 0.21 | 0.22 | 0.21 |
| D_IMF | $\sigma^2_u$ | 5.28 | 5.24 | 4.34 + 0.73 |
| | $\sigma^2_e$ | 11.51 | 11.55 | 11.72 |
| | $h^2$ | 0.32 | 0.31 | 0.3 |

**Im_777K[1]**: estimated variance components with imputed 777K SNPs; **exp_777K[2]**: estimated variance components with imputed 777K SNPs except text-mined SNPs; **exp_777K + tm_SNPs[3]**: estimated variance components when using two marker sets (exp_777K, text-mined SNPs) to different genetic variance. First genetic variance was a component of exp_777K and second was a component of text-mined SNPs.

**Table 5. Carcass traits average correlation between the GEBV and corrected phenotypic values ($y_c$) and standard error for 10-validation set.** Meat quality traits average correlation between the GEBV and corrected phenotypic values ($y_c$) and standard error for 10-validation set.

| Trait | Im_777K | exp_777K | exp_777K + tm_SNPs |
|---|---|---|---|
| CWT | 0.453 ± 0.01 | 0.449 ± 0.01 | 0.451 ± 0.01 |
| BF | 0.419 ± 0.01 | 0.413 ± 0.01 | 0.421 ± 0.01 |
| EMA | 0.423 ± 0.01 | 0.429 ± 0.01 | 0.437 ± 0.004 |
| S_SF | 0.105 ± 0.04 | 0.102 ± 0.02 | 0.129 ± 0.03 |
| D_SF | 0.121 ± 0.03 | 0.115 ± 0.04 | 0.189 ± 0.03 |
| S_IMF | 0.16 ± 0.02 | 0.15 ± 0.03 | 0.168 ± 0.02 |
| D_IMF | 0.207 ± 0.04 | 0.163 ± 0.03 | 0.225 ± 0.02 |

lipase (*LPL*) gene encodes lipoprotein lipase, which provides triglyceride-derived fatty acids to adipose tissue [52]. Fatty-acid-binding protein 4 (*FABP4*) plays a number of important roles, including fatty acid uptake, transport, and metabolism in the muscle [53].

In addition to these genes, *CAPN3*, *KCNJ11*, *DNAJA1* are also known to be associated with beef tenderness [54–56] and *FABP3*, *LEPR*, *FASN*, *DGAT1* were reported to associated with IMF in previous studies [57–59]. In the results of GO term analysis for WBSF, biological processes related to the carboxylic acid biosynthetic and metabolic processes were significant. Carboxylic acid is an organic acid that was shown in previous studies to affect beef tenderness [60, 61]. In addition, IMF related TMG showed a significant association with the regulation of lipid metabolic and biosynthetic processes. According to these biological processes, GO term results can support that WBSF, IMF–related TMG have been associated with WBSF and IMF.

## Genomic prediction

When excluding text-mined SNPs from the Im_777K marker panels, the prediction accuracy for CWT, BF, WBSF, and IMF were decreased. In a previous simulation study, a panel that excluded QTL from the 50K SNP panel showed lower accuracy than a panel that included the QTL [2]. These results indicated that text-mined SNPs may be more strongly functionally associated with QTL for CWT, BF, WBSF, and IMF and include markers in a linkage disequilibrium relationship with QTL for these traits. Fitting two GRMs constructed using exp_777K and text-mined SNPs in the GBLUP model as different random effects resulted in higher accuracy than fitting one GRM constructed using Im_777K for BF, EMA, WBSF, and IMF. These results were consistent with previous studies indicating that differentially weighted subsets of markers based on genomic features increased the predictive ability [8]. The increase in accuracy was greater in the traits related to the *longissimus dorsi* muscle than in those related to the *semimembranosus*

**Table 6. Accuracy of evenly-mined GBLUP and text-mined GBLUP.**

| Traits | exp_777k + tm_SNPs | exp_777k + em_SNPs[1] |
|---|---|---|
| CWT | 0.451 ± 0.01 | 0.471 ± 0.01 |
| BF | 0.421 ± 0.01 | 0.419 ± 0.01 |
| EMA | 0.437 ± 0.004 | 0.438 ± 0.01 |
| S_SF | 0.129 ± 0.03 | 0.099 ± 0.02 |
| D_SF | 0.189 ± 0.03 | 0.095 ± 0.02 |
| S_IMF | 0.168 ± 0.02 | 0.147 ± 0.02 |
| D_IMF | 0.225 ± 0.02 | 0.202 ± 0.03 |

**exp_777k + em_SNPs[1]**: multi-GRM GBLUP with evenly-mined SNPs and except SNPs.

muscle. One of the most important factors that can affect the accuracy of genomic prediction is linkage disequilibrium between common SNPs and QTL [7]. As selection for a specific trait proceeds, linkage disequilibrium between causal polymorphisms for that trait and other marker loci appears to be stronger [6]. As traits related to the *semimembranosus* muscle were not considered in evaluating the degree of the Hanwoo breed, the selection of these traits would not have been carried out actively. Therefore, linkage disequilibrium between QTL and other markers would be weakened, and this seemed to have been responsible for these results.

In this study, the SNPs that seemed to be related to the traits were selected by text mining, and the prediction accuracy was slightly increased when these SNPs were weighted differentially to other SNP panels. In the GBLUP method, the weights of GRMs are controlled by the lambda value ($\sigma^2_e / \sigma^2_u$). As $\sigma^2_u$ estimated by text-mined SNPs showed lower variance than estimated by exp_777K, higher lambda values were multiplied to GRM made by text-mined SNPs and this seemed to increase the prediction accuracy by giving more weight to text-mined SNPs in the model. Nevertheless, in comparisons between multi-GRM models, the accuracy of CWT and EMA decreased when tm_SNPs was used. These results may indicate that text-mined GBLUP doesn't seem to be effective in the case of traits that are more genetically affected by polygenic effect than causal variant effect. There may be limits to the conclusion that text mining can improve prediction accuracy, since text mined SNPs didn't result in a significant improvement in prediction accuracy. However, there was a slight accuracy increase for meat quality traits and GO term analysis may suggests that text mining can play a role in finding functional genes for complex traits. Therefore, attempts to incorporate text mining into genomic predictions seem valuable and further study (*i.e.*, other SNP effects weighting methods) using text mining can be expected to present the significant results [62, 63]. In addition, text mining may be used for various population or breeds, since marker selection by text mining didn't use the phenotypic or genetic information of a specific population.

## Conclusions

This study was performed to use text mining, to extract biological information from previous papers and increase the performance of genomic prediction. The results showed that text mining could be used to find genes related to specific traits because associations between each carcass and meat quality trait and TMG were identified in the results of text mining and GO term analysis. However, a word that was accidentally the same as a gene symbol but used with another meaning (*i.e.*, SST) was also mined as a text-mined gene. Therefore, it will be necessary to develop further methods of text mining that can resolve this problem. In the genomic prediction results, text-mined SNPs seemed to be in tighter linkage disequilibrium with QTL for BF, EMA, WBSF, and IMF. There may be limits to the conclusion that text mining can improve prediction accuracy, since text mined SNPs didn't result in a significant improvement in prediction accuracy. However, attempts to incorporate text mining into genomic predictions still seem valuable, and further study using text mining can be expected to present the significant results, because a slight accuracy increase for meat quality traits may suggests that text mining can play a role in finding functional genes for complex traits. In addition, text mining may be used for various population or breeds, since marker selection by text mining didn't use the phenotypic or genetic information of a specific population.

## Supporting information

**S1 Fig. The workflow of the text mining.**
(TIF)

**S1 File. SNP information used in this study.**
(ZIP)

## Acknowledgments

This study was preformed to develop new genomic selection model for carcass traits. We acknowledgment to Korea Institute for Animal Products Quality Evaluation to provide tissue sample for Genomic Reference Population.

## Author Contributions

**Conceptualization:** Seung Hwan Lee.

**Data curation:** Hak Kyo Lee, Duhak Yoon, Dajeong Lim.

**Methodology:** Hak Kyo Lee, Duhak Yoon, Dajeong Lim.

**Software:** Hyo Jun Lee, Sungbong Jang.

**Writing – original draft:** Hyo Jun Lee, Yoon Ji Chung.

**Writing – review & editing:** Dong Won Seo, Seung Hwan Lee.

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
