## [Decision Letter · Decision Letter 0]

3 Jun 2020

PONE-D-20-12127

Genome-wide identification of major genes and genomic prediction using High-Density and Text-Mined Gene-Based SNP panels in Hanwoo (Korean cattle)

PLOS ONE

Dear Dr. Lee,

Thank you for submitting your manuscript to PLOS ONE. After careful consideration, we feel that it has merit but does not fully meet PLOS ONE’s publication criteria as it currently stands. Therefore, we invite you to submit a revised version of the manuscript that addresses the points raised during the review process.

We look forward to receiving your revised manuscript.

Kind regards,

Shuhong Zhao, Ph.D

Academic Editor

PLOS ONE

Journal Requirements:

2. We note that you are reporting an analysis of a microarray, next-generation sequencing, or deep sequencing data set. PLOS requires that authors comply with field-specific standards for preparation, recording, and deposition of data in repositories appropriate to their field. Please upload these data to a stable, public repository (such as ArrayExpress, Gene Expression Omnibus (GEO), DNA Data Bank of Japan (DDBJ), NCBI GenBank, NCBI Sequence Read Archive, or EMBL Nucleotide Sequence Database (ENA)). In your revised cover letter, please provide the relevant accession numbers that may be used to access these data. For a full list of recommended repositories, see http://journals.plos.org/plosone/s/data-availability#loc-omics or http://journals.plos.org/plosone/s/data-availability#loc-sequencing.

Additional Editor Comments (if provided):

Reviewers' comments:

Reviewer's Responses to Questions

**Comments to the Author**

1. Is the manuscript technically sound, and do the data support the conclusions?

Reviewer #1: Yes

Reviewer #2: Partly

2. Has the statistical analysis been performed appropriately and rigorously? 

Reviewer #1: N/A

Reviewer #2: Yes

3. Have the authors made all data underlying the findings in their manuscript fully available?

Reviewer #1: Yes

Reviewer #2: No

4. Is the manuscript presented in an intelligible fashion and written in standard English?

Reviewer #1: No

Reviewer #2: Yes

5. Review Comments to the Author

Reviewer #1: Lee et.al conducted both GWAS and GS in a Korean cattle population. Except for the regular information of genotype and phenotype for GWAS and GS, the reported SNPs that associated with six traits were also identified by a text mining technology and used in GWAS and GS. The GWAS was carried out by a regular mixed linear model and the GS was performed by a regular GBLUP model with a single random effect or two random effects. The idea is interesting and the experimental design is fine. However, the manuscript is not well written and it is hard to judge if the statistical analyzes are done correctly. Therefore, I have the following concerns.

Major concerns:

Combined with the information from Table 1 and Table 4, we can find that the advantages of model 3 are bigger than the other two models when the number of text-mined SNPs is smaller. To remove the effect of the number of text-mined SNPs, I suggest adding one more experiment. Take CWT trait as an example, please randomly select 17,662 SNPs from all SNPs as 'fake text-mined SNPs' and test the accuracy of model 3 again and added as the fourth column in Table 4;

The published papers, which used this Korean cattle population, should not be used to identify the text-mined SNPs. This should be declared;

The GWAS was only carried out on text-mined SNPs, how about the GWAS results if using all SNPs?

The description of the GWAS model is not rigorous. For example, if the GWAS was performed by GCTA, the D should be a single text-mined SNP instead of a genotype matrix of text-mined SNPs. Please correct it and double-check the method section;

The slaughter information was added to the genomic prediction model as one of the fixed effects. What is slaughter information? Also, meat quality traits were adjusted by farmer information, what is ‘farmer information’? Was the farmer information added as fixed effects or random effects in the adjusted model. All details of the adjusted model should be clearly described.

Minor concerns:

A comma should be added in a number with more than three digits;

Why the MAF thresholds were set differently for carcass traits and meat quality traits?

The number of significant digits retained should be kept the same in each table;

What is TIAB in Table 1?

It would be more straightforward if the top genes with corresponding frequencies were shown in a Table instead of Figure1;

Some of the Chromosome IDs in Manhattan plots were overlapped.

Reviewer #2: The manuscript studied the effect of text-mined SNPs on GWAS and genomic prediction in Hanwoo. A major concern I have is that only three scenarios were compared for genomic prediction, and more scenarios should be included. Some suggestions are: 1) tm_SNPs only 2) a random subset of k SNPs from Im_777K, where k is the number of markers in exp_777K 3)fit the random subset obtained in (2) and the remaining as the two components in model 3

Other comments:

The standard error in table 4-1 is very low. Can you verify the standard error?

Why fixed effects are pre-adjusted in GWAS but not in genomic prediction?

Will GWAS using text-mined SNPs give lower power? Please add more discussion.

Line 160: Is \\beta for one candidate SNP in each test? Please correct it in line 158.

Line 209: Please correct it.

6. PLOS authors have the option to publish the peer review history of their article (what does this mean?). If published, this will include your full peer review and any attached files.

Reviewer #1: No

Reviewer #2: No

---

## [Author Response · Author response to Decision Letter 0]

13 Jul 2020

Manuscript PONE-D-20-12127R1

Dear Dr. Anita Estes

we appreciate the time and effort that you dedicated to providing feedback on our manuscript entitled “Genome-wide identification of major genes and genomic prediction using High-Density and Text-Mined Gene-Based SNP panels in Hanwoo (Korean cattle)". 

As you comment, we revised some information to comply with PLOS ONE's data sharing policy.

 1) The relevant data used in this study were named as ‘Hanwoo reference population’ (carcass trait population) and ‘Hanwoo meat quality population’. In addition, none of the authors of this study have privileges in accessing datasets. 

 2) We revised the contact for accessing the genotype and meat quality traits data to (http://www.nias.go.kr/english/sub/boardHtml.do?boardId=depintro, National Institute of Animal Science, Animal Genome & Bioinformatics Division, Tae Hun Kim, PhD, Director of Animal Genome & Bioinformatics Division, thkim63@korea.kr). Data Availability in manuscript was also revised.

 3) All SNPs position and rs-numbers used in this study were added in Supporting Information. Detailed information and accession number of each SNP can be found in Bos_taurus_UMD_3.1.1 genome information submitted in NCBI GeneBank (accession: GCA_000003055.5).

Thank you again for your efforts, and please contact us if you have any further requests.

Kind regards

SeungHwan Lee

Manuscript PONE-D-20-12127

Dear Dr. Shuhong Zhao

Thank you for giving us the opportunity to submit a revised manuscript ‘Genome-wide identification of major genes and genomic prediction using High-Density and Text-Mined Gene-Based SNP panels in Hanwoo (Korean cattle)’ to PLOS ONE. We appreciate you for careful and thorough reading of our manuscript and valuable comment to our manuscript.

As you comment, the reference citation and figure file naming were changed following the PLOS ONE style templates. In addition, captions for Supporting Information files were added at the end of our revised manuscript.

We totally agree that the data sharing is important for reproducibility, replication, and validation of research. However, some of our data cannot be shared due to legal restrictions on data sharing. All the data-set (Phenotypes and genotypes for carcass traits) used in this study was provided by BioGreen 21 Program (Molecular Breeding Program) of National Institute of Animal Science, RDA. The carcass traits can be obtained at public web site (https://mtrace.go.kr/). Request for Genotype and meat quality traits data can be made to Korea National Institute of Animal Science (https://www.nias.go.kr:3443/front/main.do) at (lim.dj@korea.kr). We apologize for our situation and detailed data availability statement was added at the revised manuscript.

Response to Reviewers

Reviewer #1: Lee et.al conducted both GWAS and GS in a Korean cattle population. Except for the regular information of genotype and phenotype for GWAS and GS, the reported SNPs that associated with six traits were also identified by a text mining technology and used in GWAS and GS. The GWAS was carried out by a regular mixed linear model and the GS was performed by a regular GBLUP model with a single random effect or two random effects. The idea is interesting and the experimental design is fine. However, the manuscript is not well written and it is hard to judge if the statistical analyzes are done correctly. Therefore, I have the following concerns. 

Response: we appreciate the time and effort that you dedicated to providing feedback on our manuscript. We have incorporated most of your suggestions. Your insightful comment made our papers more valuable. We written in red for a point-by-point response to your concerns. Thank you.

Major concerns:

Combined with the information from Table 1 and Table 4, we can find that the advantages of model 3 are bigger than the other two models when the number of text-mined SNPs is smaller. To remove the effect of the number of text-mined SNPs, I suggest adding one more experiment. Take CWT trait as an example, please randomly select 17,662 SNPs from all SNPs as 'fake text-mined SNPs' and test the accuracy of model 3 again and added as the fourth column in Table 4;

Response: We appreciate the reviewer’s feedback, we agree that the analysis using random sets should be added. However, in the model 3, it is important to identify how independent between called (text-mined SNPs) and non-called (exp SNPs) regions are. If the SNPs are evenly (or randomly) extracted, both SNP set (called, non-called) will exhibit similar patterns due to LD. In the linear model, Fitting two highly correlated GRMs will be less accurate or similar to fitting the GRM using the whole SNPs. Therefore, we thought that using the evenly extracted SNPs to model 3 will not be much different from the results we want to show in model 1. The table below shows the results of Model 3 using evenly extracted SNPs and text-mined SNPs. the second table is the correlation between GRMs constructed with each SNP set.

　 text_mined + excepted evenly_extracted + excepted Im_777K

CWT 0.451 ± 0.01 0.4707 ± 0.01 0.453 ± 0.01

BF 0.421 ± 0.01 0.4187 ± 0.01 0.419 ± 0.01

EMA 0.437 ± 0.004 0.4376 ± 0.01 0.423 ± 0.01

S_SF 0.129 ± 0.03 0.0992 ± 0.02 0.105 ± 0.04

D_SF 0.189 ± 0.03 0.0954 ± 0.02 0.121 ± 0.03

S_IMF 0.168 ± 0.02 0.1471 ± 0.02 0.16 ± 0.02

D_IMF 0.225 ± 0.02 0.202 ± 0.03 0.207 ± 0.04

 text_mined + excepted evenly_extracted + excepted

CWT 0.586 0.784

BF 0.576 0.596

EMA 0.578 0.642

WBSF 0.71 0.981

IMF 0.908 0.997

The published papers, which used this Korean cattle population, should not be used to identify the text-mined SNPs. This should be declared;

Response: Thank you for pointing this out. There was no published paper using the Carcass trait population (n = 12,635) when conducting text-mining (sep, 2019). There is one published paper using part of the Meat quality population (n = 1,039), however the abstract of this paper does not contain the Gene symbol. Therefore, we can declare that published papers using this Korean cattle dataset were not included in text-mining.

The GWAS was only carried out on text-mined SNPs, how about the GWAS results if using all SNPs? 

Response: Indeed, we had expected that GWAS with text-mined SNPs would be able to identify the distribution of QTLs whose effect disappeared during the selection process. However, these results were not much different with the all SNPs results. It seems that text-mined SNPs have not been able to filter out common SNPs that have little or no effect but appear to be high due to the LD, because the text-mined SNPs were distributed throughout the whole genomes. The figures below show the GWAS results using the all SNPs and the simulation results for the QTL effect chainging with selection.

G0 (no selection)

G10 (after 10 generation with selection)

The description of the GWAS model is not rigorous. For example, if the GWAS was performed by GCTA, the D should be a single text-mined SNP instead of a genotype matrix of text-mined SNPs. Please correct it and double-check the method section;

Response: We apologize for the confusion. In this study, single marker linear mixed model was used for GWAS. Therefore, D must be genotype vector for each SNP, not genotype matrix. We corrected GWAS description (line 160) to ‘D is a vector of genotype of the candidate SNPs recorded as 0,1, or2’. In addition, in single marker liner mixed model, beta of each model means additive effect of single marker. Therefore, we also revised description (line 161) to ‘beta is the additive effect of the candidate SNPs’ 

The slaughter information was added to the genomic prediction model as one of the fixed effects. What is slaughter information? Also, meat quality traits were adjusted by farmer information, what is ‘farmer information’? Was the farmer information added as fixed effects or random effects in the adjusted model. All details of the adjusted model should be clearly described.

Response: Thank you for pointing this confusion out. In this study, the fixed effect (birth information, slaughter information) used in common for both population (carcass, meat quality) mean birth month, birth year, slaughter month, and slaughter year. We have the slaughter place information for carcass traits population, therefore we used this information as fixed effect for carcass traits. In case of meat quality population, we use the farmer information as fixed information, because we had the name of the owner for the slaughter steer (farmer information). To clarify these, we rewrote (line 177) to ‘b is the vector of fixed effects, which included growing site, birth month, birth year, slaughter month, slaughter year, and slaughter age as covariates for all traits. In addition, the carcass traits included slaughter place and sex, while the meat quality trait included farmer information (the owner’s name of steers).’

Minor concerns:

A comma should be added in a number with more than three digits.

Response: we revised all number with more than three digits.

Why the MAF thresholds were set differently for carcass traits and meat quality traits?

Response: You have raised an important point, we wanted to remove SNPs that less than 10 individuals had, because SNPs that too few individuals had are at risk of overestimating effects in GWAS or Genomic prediction. Since 10 individuals were equivalent to 0.001 of the carcass population and 0.01 of the meat quality population, different MAF thresholds were set for two population.

The number of significant digits retained should be kept the same in each table

Response: The significant digits and Gene ratio shown in table2 (table3 in the revised manuscript) were modified to show only one decimal point.

What is TIAB in Table 1?

Response: [TIAB] is one of the search options for Pubmed search engine. With [TIAB] option, only the papers which included words and numbers in a citation’s title, collection title, abstract, other abstract and keywords were search in Pubmed database. Other options and descriptions for Pubmed engine can be found in ‘https://www.ncbi.nlm.nih.gov/books/NBK3827/’.

It would be more straightforward if the top genes with corresponding frequencies were shown in a Table instead of Figure1.

Response: Thanks for pointing out. As you commented, the contents of all Figure 1 have been changed to Table2 for more tidy result delivery.

Some of the Chromosome IDs in Manhattan plots were overlapped.

Response: The revised pictures for the overlapping Chromosome IDs were re-attached.

Reviewer #2: The manuscript studied the effect of text-mined SNPs on GWAS and genomic prediction in Hanwoo. A major concern I have is that only three scenarios were compared for genomic prediction, and more scenarios should be included. Some suggestions are: 1) tm_SNPs only 2) a random subset of k SNPs from Im_777K, where k is the number of markers in exp_777K 3)fit the random subset obtained in (2) and the remaining as the two components in model 3

Response: We appreciate the reviewer’s comment, we agree that the analysis using tm_SNPs only and random sets should be added. In scenarios using only tm_SNPs, prediction accuracy was lower than other scenarios. There seem to be two reason for these results. 1) there are some polygenic effect missing, i.e. interaction between markers, since only tm_SNP is used for genomic prediction. 2) text-mining didn’t identify all causal variant. We didn’t wrote these tm_SNPs results in the manuscript, because these results were thought to be inevitable in a genomic prediction using low density marker set. 

In addition, we had conducted the genomic prediction fitting evenly selected SNPs to model 3. However, the independence of the two GRM (tm_SNP + exp_SNP) is important for statistical power of model 3 because the collinearity of two GRMs could make model weaken. Therefore, we thought that using the evenly (or randomly) extracted SNPs to model 3 will not be much different from the results we want to show in model 1. The table below shows the results of Model 3 using evenly extracted SNPs and text-mined SNPs. the second table is the correlation between GRMs constructed with each SNP set.

 

　 text_mined + excepted evenly_extracted + excepted Im_777K

CWT 0.451 ± 0.01 0.4707 ± 0.01 0.453 ± 0.01

BF 0.421 ± 0.01 0.4187 ± 0.01 0.419 ± 0.01

EMA 0.437 ± 0.004 0.4376 ± 0.01 0.423 ± 0.01

S_SF 0.129 ± 0.03 0.0992 ± 0.02 0.105 ± 0.04

D_SF 0.189 ± 0.03 0.0954 ± 0.02 0.121 ± 0.03

S_IMF 0.168 ± 0.02 0.1471 ± 0.02 0.16 ± 0.02

D_IMF 0.225 ± 0.02 0.202 ± 0.03 0.207 ± 0.04

 text_mined + excepted evenly_extracted + excepted

CWT 0.586 0.784

BF 0.576 0.596

EMA 0.578 0.642

WBSF 0.71 0.981

IMF 0.908 0.997

Other comments:

The standard error in table 4-1 is very low. Can you verify the standard error?

Response: We calculated standard error by

1/√K sd{〖CV〗_1 (〖cor〗^(-(1) ) ),…〖CV〗_K (〖cor〗^(-(K)))}

Where, 〖CV〗_n (〖cor〗^(-(n) ) ) is Pearson's correlation of test set of Validation N. In this study, we use 10 – fold validation for calculating accuracy. Therefore, we calculated the standard deviation of {〖CV〗_1 (〖cor〗^(-(1) ) ),…〖CV〗_10 (〖cor〗^(-(10)))}, and the value divided by √10 (1/√10 sd{〖CV〗_1 (〖cor〗^(-(1) ) ),…〖CV〗_10 (〖cor〗^(-(10)))}) was taken as the standard error.

Why fixed effects are pre-adjusted in GWAS but not in genomic prediction?

Response: We apologize for the confusion. and thank you for pointing this problem out. The GCTA program conduct association study by two steps, in order to improve computational efficiency. step1) the phenotype was adjusted by the mean and covariates (fixed effect), step2) the adjusted phenotype will subsequently be used for testing SNP association. Therefore, we pre-adjusted the phenotype to skip the step1. On the other hand, we didn’t pre-adjusted phenotype in genomic prediction, because the BLUPF90 program fits all fixed effects in one model. 

If SNPs are correlated with the fixed effects, pre-adjusting the phenotype by the fixed effects will probably cause overestimation of SNP effect. However, in this study, the environmental effect was used as a fixed effect. In addition, since SNP has not been mutated during the lifetime, there could be no interaction with the environment. Therefore, there are no difference between the two method (pre-adjusted or not).

Will GWAS using text-mined SNPs give lower power? Please add more discussion.

Response: In the individual selection processes for breeding, it is difficult to identify the QTL by GWAS as the effect of QTL decreases. Therefore, we expected that text-mining could filter out common-SNPs whose effect is overestimated by LD with causal variants. However, the GWAS using text-mined SNPs did not seem to significantly differ from the results using 777k SNPs. these results seem to be because text-mined SNPs were distributed throughout the whole genomes. Nevertheless, GWAS using text-mined SNPs still seems valuable in terms of identifying the association between causal variants candidate SNPs and phenotypes. The figures below show the GWAS results using the 777k SNPs and the simulation results for the QTL effect chainging with selection.

G0 (no selection)

G10 (after 10 generation with selection)

Line 160: Is \\beta for one candidate SNP in each test? Please correct it in line 158.

Response: We apologize for the confusion. In this study, single marker linear mixed model was used for GWAS. Therefore, beta must be one value for each SNP, not vector of the effect. We revised GWAS description (line 161) to ‘beta is the additive effect of the candidate SNPs’. Thank you for pointing out this problem. 

Line 209: Please correct it.

Response: We apologize for the miswriting, and We thank the reviewer for this comment. We wanted to present the equation for calculating the GEBV of the reference and validation set in a mixed model. However, there were some miswriting (i.e G is inverse of the GRM, not GRM). therefore, we removed this equation and added two new equation in (line 209, 215). First equation is linear mixed model with single random effect for the model1 and model2, second equation is mixed model with two random effect for the model3. We thought this way will more clearly explain our prediction models.

---

## [Decision Letter · Decision Letter 1]

14 Aug 2020

PONE-D-20-12127R1

Genome-wide identification of major genes and genomic prediction using High-Density and Text-Mined Gene-Based SNP panels in Hanwoo (Korean cattle)

PLOS ONE

Dear Dr. Lee,

Thank you for submitting your manuscript to PLOS ONE. After careful consideration, we feel that it has merit but does not fully meet PLOS ONE’s publication criteria as it currently stands. Therefore, we invite you to submit a revised version of the manuscript that addresses the points raised during the review process.

We look forward to receiving your revised manuscript.

Kind regards,

Shuhong Zhao, Ph.D

Academic Editor

PLOS ONE

Reviewers' comments:

Reviewer's Responses to Questions

**Comments to the Author**

1. If the authors have adequately addressed your comments raised in a previous round of review and you feel that this manuscript is now acceptable for publication, you may indicate that here to bypass the “Comments to the Author” section, enter your conflict of interest statement in the “Confidential to Editor” section, and submit your "Accept" recommendation.

Reviewer #1: All comments have been addressed

Reviewer #2: (No Response)

2. Is the manuscript technically sound, and do the data support the conclusions?

Reviewer #1: Yes

Reviewer #2: Partly

3. Has the statistical analysis been performed appropriately and rigorously? 

Reviewer #1: Yes

Reviewer #2: Yes

4. Have the authors made all data underlying the findings in their manuscript fully available?

Reviewer #1: (No Response)

Reviewer #2: No

5. Is the manuscript presented in an intelligible fashion and written in standard English?

Reviewer #1: Yes

Reviewer #2: Yes

6. Review Comments to the Author

Reviewer #1: The authors have followed my recommendations to improve the manuscript. I suggest to add the information of tables and figures in point by point response to the manuscript and discuss the newly added results as well.

Reviewer #2: In author's reply, authors mentioned a potential reason for the results that prediction using tm_SNP only gives lower accuracy is "text-mining didn’t identify all causal variant.". If this is the reason, GWAS using tm_SNPs may cause lower power because causal variants excluded from tm_SNPs are not tested. In terms of prediction, significant differences are rarely observed, and it looks like a significant difference is observed for CWT, but "text_mined + excepted" is worse than "evenly_extracted + excepted". I would recommend authors to run more cross validation to further verify their conclusion. A paired t test may also be considered.

7. PLOS authors have the option to publish the peer review history of their article (what does this mean?). If published, this will include your full peer review and any attached files.

Reviewer #1: No

Reviewer #2: No

---

## [Author Response · Author response to Decision Letter 1]

24 Aug 2020

Reviewer #1: The authors have followed my recommendations to improve the manuscript. I suggest to add the information of tables and figures in point by point response to the manuscript and discuss the newly added results as well.

Thank you for this suggestion. As suggested by the reviewer, we added the evenly_SNPs + except results at table 6. 

Table 6. Accuracy of evenly-mined GBLUP and text-mined GBLUP

Traits exp_777k + tm_SNPs exp_777k + em_SNPs1

CWT 0.451 ± 0.01 0.471 ± 0.01

BF 0.421 ± 0.01 0.419 ± 0.01

EMA 0.437 ± 0.004 0.438 ± 0.01

S_SF 0.129 ± 0.03 0.099 ± 0.02

D_SF 0.189 ± 0.03 0.095 ± 0.02

S_IMF 0.168 ± 0.02 0.147 ± 0.02

D_IMF 0.225 ± 0.02 0.202 ± 0.03

exp_777k + em_SNPs1: multi-GRM GBLUP with evenly-mined SNPs and except SNPs.

These results have shown, in CWT, em_SNPs can make more accurate prediction than tm_SNPs. As with the results using whole SNPs, The polygenic characteristic of traits seem to make these results. These results may indicate that text-mined GBLUP doesn’t seem to be effective in the case of traits that are more genetically affected by polygenic effect than causal variant effect. We also added this finding in the discussion (line 398). Thank you again for your comment.

Reviewer #2: In author's reply, authors mentioned a potential reason for the results that prediction using tm_SNP only gives lower accuracy is "text-mining didn’t identify all causal variant.". If this is the reason, GWAS using tm_SNPs may cause lower power because causal variants excluded from tm_SNPs are not tested. In terms of prediction, significant differences are rarely observed, and it looks like a significant difference is observed for CWT, but "text_mined + excepted" is worse than "evenly_extracted + excepted". I would recommend authors to run more cross validation to further verify their conclusion. A paired t test may also be considered.

Thank you for pointing this out. We totally agree that if text-mining can’t identify all causal variant, the GWAS using tm_SNPs may have lower power than using whole SNPs. However, the lower accuracy of tm_SNPs only seems to be more affected by the loss of polygenic effect (e.g gene x gene, gene x SNPs etc) than missing the causal variant. Even though the text-mining misses a small number of causal variants (variant recall will be decreased.), it still seems good way to filter out a large number of common SNPs (variant precision will be increased.!). According to this description, this method may have lower power to identifying the genetic architecture of target traits. However, considering the ability to remove the SNPs that make noise to GWAS with LD, it seems to be a good method for finding true causal variants of target traits.

In terms of the genomic prediction results, as you commented on, only CWT and D_SF showed significant differences (by t-test). The reason why em_SNPs showed higher accuracy in CWT is seem to be because CWT is more affected by polygenic effect than causal variant effect due to the characteristic of quantitative traits (or carcass trait). We added these results at table 6, and discussion for these results also added (line 398). Thank you again for the time and effort that you dedicated to providing feedback on our manuscript.

Table 6. Accuracy of evenly-mined GBLUP and text-mined GBLUP

Traits exp_777k + tm_SNPs exp_777k + em_SNPs1

CWT 0.451 ± 0.01 0.471 ± 0.01

BF 0.421 ± 0.01 0.419 ± 0.01

EMA 0.437 ± 0.004 0.438 ± 0.01

S_SF 0.129 ± 0.03 0.099 ± 0.02

D_SF 0.189 ± 0.03 0.095 ± 0.02

S_IMF 0.168 ± 0.02 0.147 ± 0.02

D_IMF 0.225 ± 0.02 0.202 ± 0.03

exp_777k + em_SNPs1: multi-GRM GBLUP with evenly-mined SNPs and except SNPs.

---

## [Decision Letter · Decision Letter 2]

30 Sep 2020

PONE-D-20-12127R2

Genome-wide identification of major genes and genomic prediction using High-Density and Text-Mined Gene-Based SNP panels in Hanwoo (Korean cattle)

PLOS ONE

Dear Dr. Lee,

Thank you for submitting your manuscript to PLOS ONE. After careful consideration, we feel that it has merit but does not fully meet PLOS ONE’s publication criteria as it currently stands. Therefore, we invite you to submit a revised version of the manuscript that addresses the points raised during the review process.

We look forward to receiving your revised manuscript.

Kind regards,

Shuhong Zhao, Ph.D

Academic Editor

PLOS ONE

Reviewers' comments:

Reviewer's Responses to Questions

**Comments to the Author**

1. If the authors have adequately addressed your comments raised in a previous round of review and you feel that this manuscript is now acceptable for publication, you may indicate that here to bypass the “Comments to the Author” section, enter your conflict of interest statement in the “Confidential to Editor” section, and submit your "Accept" recommendation.

Reviewer #1: All comments have been addressed

Reviewer #2: All comments have been addressed

2. Is the manuscript technically sound, and do the data support the conclusions?

Reviewer #1: Yes

Reviewer #2: Yes

3. Has the statistical analysis been performed appropriately and rigorously? 

Reviewer #1: Yes

Reviewer #2: Yes

4. Have the authors made all data underlying the findings in their manuscript fully available?

Reviewer #1: Yes

Reviewer #2: Yes

5. Is the manuscript presented in an intelligible fashion and written in standard English?

Reviewer #1: Yes

Reviewer #2: Yes

6. Review Comments to the Author

Reviewer #1: The manuscript has been improved and my previous concerns were addressed. I have one more comment:

It is very useful but also very challenge to weight a set of genetic markers to improve prediction accuracy. A recent study developed a new method to improve the prediction accuracy by weighting the contributions of genetic markers in the prediction model (Lilin Yin et. al, Genome Biology, 2020). It should be nice if this could be discussed to compare with your strategy.

Reviewer #2: I have no further comments.

7. PLOS authors have the option to publish the peer review history of their article (what does this mean?). If published, this will include your full peer review and any attached files.

Reviewer #1: No

Reviewer #2: No

---

## [Author Response · Author response to Decision Letter 2]

4 Oct 2020

Reviewer #1: The manuscript has been improved and my previous concerns were addressed. I have one more comment:

It is very useful but also very challenge to weight a set of genetic markers to improve prediction accuracy. A recent study developed a new method to improve the prediction accuracy by weighting the contributions of genetic markers in the prediction model (Lilin Yin et. al, Genome Biology, 2020). It should be nice if this could be discussed to compare with your strategy.

Response: Thank you for introducing a good method. We also agreed that weighting genetic markers is a challenge and further study is required. Therefore, we revised line 405 included at discussion to "Therefore, attempts to incorporate text mining into genomic predictions seem valuable and further study (i.e., other SNP effects weighting methods) using text mining can be expected to present the significant results.". In addition, the paper you recommended and (Wang H et al., Genetics Research, 2012) paper were referenced at the end of the sentence.

Reviewer #2: I have no further comments.

Response: Thank you for the time and effort you made during the whole review process.

---

## [Editor Report · Decision Letter 3]

22 Oct 2020

Genome-wide identification of major genes and genomic prediction using High-Density and Text-Mined Gene-Based SNP panels in Hanwoo (Korean cattle)

PONE-D-20-12127R3

Dear Dr. Lee,

We’re pleased to inform you that your manuscript has been judged scientifically suitable for publication and will be formally accepted for publication once it meets all outstanding technical requirements.

Kind regards,

Shuhong Zhao, Ph.D

Academic Editor

PLOS ONE
---

## [Editor Report · Acceptance letter]

29 Oct 2020

PONE-D-20-12127R3 

Genome-wide identification of major genes and genomic prediction using High-Density and Text-Mined Gene-Based SNP panels in Hanwoo (Korean cattle) 

Dear Dr. Lee:

I'm pleased to inform you that your manuscript has been deemed suitable for publication in PLOS ONE. Congratulations! Your manuscript is now with our production department. 

Kind regards, 

on behalf of

Mrs. Shuhong Zhao 

Academic Editor

PLOS ONE